# Targeting operational regimes of interest in recurrent neural networks

**Pierre Ekelmans**[1,2], **Nataliya Kraynyukova**[1,3], **Tatjana Tchumatchenko**[1,3,4] *

**1** Theory of Neural Dynamics group, Max Planck Institute for Brain Research, Frankfurt am Main, Germany, **2** Frankfurt Institute for Advanced Studies, Frankfurt am Main, Germany, **3** Institute of Experimental Epileptology and Cognition Research, Life and Brain Center, Universitätsklinikum Bonn, Bonn, Germany, **4** Institute of physiological chemistry, Medical center of the Johannes Gutenberg-University Mainz, Mainz, Germany

* tatjana.tchumatchenko@uni-mainz.de

**Data Availability Statement:** All files can be found in this repository: https://gitlab.rlp.net/braincodepublished/neuronal-network-simulations-June-2021-/-/tree/Ekelmans_et_al_2023.

## Abstract

Neural computations emerge from local recurrent neural circuits or computational units such as cortical columns that comprise hundreds to a few thousand neurons. Continuous progress in connectomics, electrophysiology, and calcium imaging require tractable spiking network models that can consistently incorporate new information about the network structure and reproduce the recorded neural activity features. However, for spiking networks, it is challenging to predict which connectivity configurations and neural properties can generate fundamental operational states and specific experimentally reported nonlinear cortical computations. Theoretical descriptions for the computational state of cortical spiking circuits are diverse, including the balanced state where excitatory and inhibitory inputs balance almost perfectly or the inhibition stabilized state (ISN) where the excitatory part of the circuit is unstable. It remains an open question whether these states can co-exist with experimentally reported nonlinear computations and whether they can be recovered in biologically realistic implementations of spiking networks. Here, we show how to identify spiking network connectivity patterns underlying diverse nonlinear computations such as XOR, bistability, inhibitory stabilization, supersaturation, and persistent activity. We establish a mapping between the stabilized supralinear network (SSN) and spiking activity which allows us to pinpoint the location in parameter space where these activity regimes occur. Notably, we find that biologically-sized spiking networks can have irregular asynchronous activity that does not require strong excitation-inhibition balance or large feedforward input and we show that the dynamic firing rate trajectories in spiking networks can be precisely targeted without error-driven training algorithms.

## Author summary

Biological neural networks must be able to execute diverse nonlinear operations on signals in order to perform complex information processing. While nonlinear transformations have been observed experimentally or in specific theoretical models, a comprehensive theory linking the parameters of a network of spiking neurons to its computations is still

**Funding:** This work was supported by the Max Planck Society, University of Bonn Medical Center, University of Mainz Medical Center, German Research Foundation (CRC 1080, SPP2041 to T.T.) and the LOEWE Center for Multiscale Modelling in Life Sciences funded by the federal state of Hessen. The funders had no role in study design, data collection and analysis, decision to publish, or preparation of the manuscript.

**Competing interests:** The authors have declared that no competing interests exist.

lacking. We show that spiking networks can be accurately approximated with a mathematically tractable model, the Stabilized Supralinear Network. Using the mapping we derived between these two frameworks, we show that spiking networks have a rich repertoire of nonlinear regimes at their disposal and link the existence of such regimes to precise conditions on parameters. Notably, we show that classical excitatory-inhibitory networks of leaky integrate-and-fire neurons support nonlinear transformations without the need for synaptic plasticity, intricate wiring diagrams or a complex system of different cell types. The capacity of a network to reliably perform such operations has profound functional implications as they can be the basis permitting the execution of complex computations.

## Introduction

Layered or columnar neuronal structures consisting of hundreds to thousands of neurons constitute local computational blocks in the mammalian cortex. Each computational block has its particular size and connectivity rules, which determine its dynamics and computational repertoire. Therefore, understanding the computational regimes of recurrent networks with sizes ranging from hundreds to millions of neurons and diverse connectivity patterns is essential to explain the emergence of cognitive functions and behavior. While powerful mathematical theories can operate at opposite scales, from a small number of neurons generating a particular activity pattern [1] to the limit of infinitely large networks [2], it is currently challenging to quantitatively predict the activity of biologically-sized spiking neural networks whose size lie between these two limits. However, growing amount of datasets containing activity recordings of thousands of neurons require theories that can make mathematically tractable, quantitative, and experimentally relevant predictions for the sizes of spiking networks reported for local cortical circuits [3–5].

Here, we study the activity regimes of spiking networks whose sizes range from a few hundred to thousands of neurons. Many parameters describing spiking neurons and their intracortical connections have recently been measured across cortical cell types [6], and detailed numerical network simulations have been put forward [7]. However, it is challenging to interpret and generalize spiking network simulations because network dynamics depend strongly on multidimensional parameter settings, while experimentally reported parameters vary across broad ranges [8–12].

An alternative to detailed numerical simulations is provided by population rate models [13–15] which describe the average activity of neurons in each population and can relate the activity in a complex neural network to a few underlying parameters characterizing the connectivity and the properties of neurons. Many such models exist which differ by the features considered and the level of complexity of their mathematical formulation. Consequently, multiple rate models can precisely predict the dynamics of neural networks by employing mathematically exact descriptions of the neuronal response to input [14, 16–18] or by considering finite-size deviations from a mean-field approach such as the effect of correlations and fluctuations [17, 19, 20]. Yet, this high fidelity comes at the expense of mathematical tractability. The most accurate models are difficult to manipulate, which makes it challenging to predict theoretically the computational regime of a network from its parameter configuration.

On the other side of the complexity spectrum is the balanced state framework [2, 21], which provides a powerful and mathematically tractable model in the limit of infinitely large networks. Its biological correlate is the experimentally reported strong balance between excitatory

and inhibitory synaptic currents [22–24] and results in asynchronous irregular spiking activity [25]. However, the computational hallmarks of the balanced network limit, including response linearity and strong feedforward connections, are not consistent with a set of experimentally reported non-linear responses across cortical areas [26] and reports of weak feedforward inputs [27]. Furthermore, the existence of a stable balanced solution imposes strict conditions on the connectivity configuration [21, 28] which are not guaranteed to be met in biological neural networks.

Finally, the stabilized supralinear network model (SSN) [29] is a phenomenological rate model which does not come with such strict restrictions on parameters. It is based on a supra-linear power law as a transfer function. The advantage of the SSN framework is that its activity states can be characterized analytically [15, 30] and it can reproduce a variety of nonlinear cortical responses in the realistic range of firing rates observed *in vivo* [29].

Could the SSN model provide a tractable framework to predict and quantify the activity regimes in biologically-sized spiking networks for arbitrary connectivity configurations? Here, we show that the SSN model can be used to predict diverse nonlinear responses such as super-saturation, bistable activity, and inhibition stabilized regimes [31, 32] in spiking networks. We propose a mapping between the high-dimensional parameter space of the leaky-integrate-and-fire (LIF) network of spiking neurons to the SSN model which results in a mathematically tractable model which can be manipulated analytically. With this work, we provide an easy-to-implement analytical approximation of the spiking network which relies on the SSN model. This allows us to predict the computational regime of a network of LIF neurons based on its parameters. We show how this mapping can be used to design a neural network to target a desired activity trajectory or operational regime of interest without network training. This approach can be used to generate specific nonlinear functions (eg: XOR gate). Furthermore, we find that not only biologically-sized but also much larger spiking networks can have a complex nonlinear behavior that can be more accurately described with the SSN framework than the balanced state theory.

## Results

Our goal is to understand how neural circuits comprising a few thousand neurons organize their spiking activity. We want to predict whether specific nonlinear computations can occur in these networks and pinpoint their location in the multidimensional parameter space spanned by recurrent connectivity and input weights. We choose the size of the networks to be 4000 neurons, which is biologically plausible for local cortical circuits (S1 Text). Neurons belong to one of two homogeneous populations, Excitatory (E) and Inhibitory (I). We restrict our analysis to the the range of 0–10 Hz, which is consistent with sustained population-averaged activity levels reported *in vivo* [33–40]. We choose the strength and probability of synaptic connections to be within the same order of magnitude as the values reported by the database of the Allen Institute for the visual cortex area V1 in mice [6] (S1 Text). To model cortical activity, we use the leaky-integrate-and-fire (LIF) model (see Materials and methods), which represents a useful description of cortical neurons both *in vivo* and *in vitro* [41]. To predict spiking network activity regimes, we map the mean activity of spiking networks to a rate-based 2D SSN model.

### Approximating spiking network activity with the SSN model

Our starting point is the observation that a power-law function can accurately describe the F-I curve of a single LIF neuron in response to white noise input across different membrane time constants $\tau$ and input noise values $\sigma$ (see Fig 1B and 1C). Throughout this study, we are

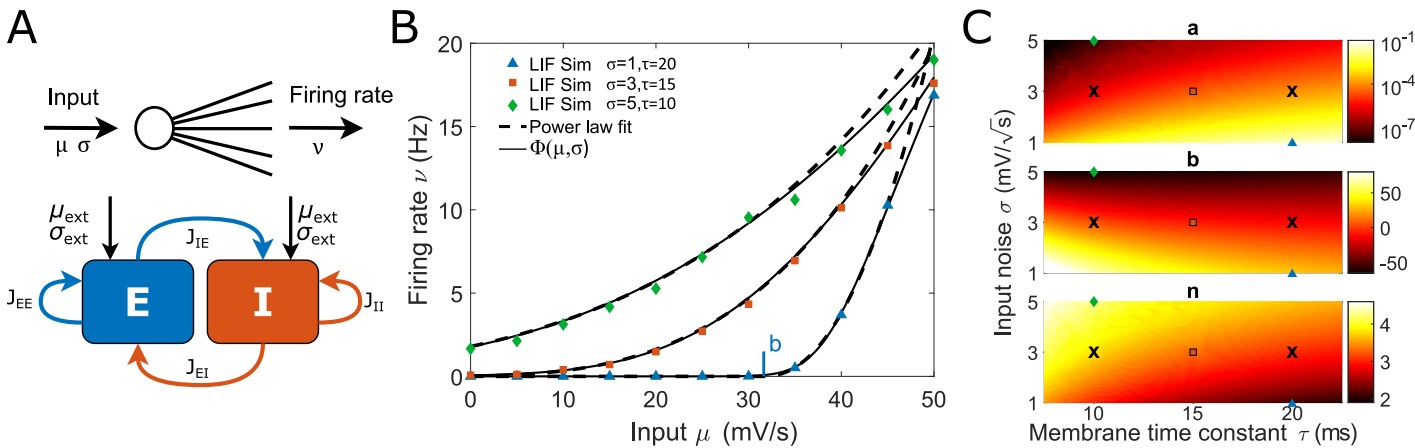

**Fig 1. Spiking neurons can be quantitatively described by a supralinear power law for low activity.** (A, top) Schematic representation of the F-I transfer function of a neuron. (A, bottom) Architecture of the recurrent Excitatory-Inhibitory network. (B) Neuronal firing rate as a function of input for different input noise $\sigma$ and membrane time constant $\tau$. The power-law approximation (Eq 1) accurately aligns with the LIF neuron simulation and $\Phi$ transfer function Eq 9 for low firing rates. Note that the power-law fit is only applied in the range of $v < 10$ Hz, and diverges beyond this range. The vertical mark denotes the power-law parameter $b$ for one of the curves ($b$ is negative for the other two curves). (C) The power-law parameters depend on input noise $\sigma$ and the membrane time constant $\tau$. The two crosses indicate the parameter regimes we use for the excitatory ($\tau_E = 20$ ms) and inhibitory ($\tau_I = 10$ ms) neurons in recurrent networks. Other symbols indicate the parameters $\tau$ and $\sigma$ used in B. The fit is obtained with the least squares method. Power-law parameters are listed in Table D in S1 Text.

interested in the firing rate range from zero to approximately 10 Hz which has been reported *in vivo* across many brain areas [33–40]. To this end, we fit the low firing rate regime (0–10 Hz) of the F-I curve of a LIF neuron given by the Ricciardi transfer function $\Phi$ Eq 9 [42] using the threshold power-law function of the form

$$v = a(\mu - b)_+^n. \tag{1}$$

Where $v$ is the firing rate, $\mu$ is the input to the neuron and $(x)_+ = \max\{x, 0\}$. The constants $a$, $b$, and $n$, which are obtained by fitting the $\Phi$ function Eq 9, characterize the power-law approximation with a scaling pre-factor $a$, an input threshold $b$ upon which the neuron starts firing, and an exponent $n$. The power-law exponent $n$ in our approximation varies between 2 and 4, which is consistent with the biologically reported range [38, 43].

We connect the individual LIF neurons into a recurrent network of excitatory (E) and inhibitory (I) neurons (Fig 1A). We assume that E and I neurons differ in their membrane time constants ($\tau_E = 20$ ms, $\tau_I = 10$ ms, black crosses in Fig 1C) consistently with experimental reports [6]. We note that the input to a neuron in a recurrent network, which is a superposition of postsynaptic potentials (PSPs), is equivalent to an Ornstein Uhlenbeck process or white noise if the number of incoming PSPs is sufficiently large and the activity is temporally uncorrelated or Poissonian [44, 45].

To describe the activity of the E and I populations, we use the power-law approximation of the single-neuron transfer function (Eq 1). In the mean field approximation, the average firing rate of each population is given by a system of equations equivalent to the SSN [29]

$$\begin{aligned}
\tau_{P_E} \frac{dv_E}{dt} &= -v_E + a_E(\mu_E - b_E)_+^{n_E} \\
\tau_{P_I} \frac{dv_I}{dt} &= -v_I + a_I(\mu_I - b_I)_+^{n_I}.
\end{aligned} \tag{2}$$

Where $v_X$, $X \in \{E, I\}$ are the firing rates of the two populations, the parameters $a_X$, $b_X$, and $n_X$ are given by the F-I curve fit of E and I neurons (black crosses in Fig 1C), and $\mu_X$ is the total

input to the neurons in each population. In a recurrent E-I network, $\mu_X$ is the sum of recurrent ($J_{XE}\nu_E - J_{XI}\nu_I$, see Eq 12) and feedforward ($\mu_{extX}$) inputs to each population. The population time constants $\tau_{P_X}$ characterize how fast the firing rate of each population evolves. *In vivo* cortical networks have been shown to respond to sudden stimulation with a transient (or *onset*) response that has a time scale of approximately 20 ms [46]. Meanwhile, *in vivo* recordings of multiple cortical areas have reported autocorrelation on a much slower timescales, of the order of hundreds of ms [47–49]. This suggests that the timescales over which biological networks typically evolve *in vivo* is driven by extrinsic factors such as changes in feedforward input, rather than the intrinsic timescales $\tau_{P_X}$ which emerge from synaptic and neuronal variables. Therefore, in this work we restrict our analysis to the study of firing rates dynamics which evolve on sufficiently slow timescales to assume that the network operates at the equilibrium state. While the study of fast or transient dynamics would necessitate to consider the temporal characteristics of the network Eq 2, the steady states of the SSN are described by

$$
\begin{aligned}
\nu_E &= a_E(J_{EE}\nu_E - J_{EI}\nu_I + \mu_{ext} - b_E)_+^{n_E} \\
\nu_I &= a_I(J_{IE}\nu_E - J_{II}\nu_I + r\mu_{ext} - b_I)_+^{n_I}.
\end{aligned}
\tag{3}
$$

Here $r$ is the ratio of the external inputs to the I and E populations $r = \mu_{extI}/\mu_{extE}$, which allows for the simplified notation: $\mu_{ext} = \mu_{extE}$ and $r\mu_{ext} = \mu_{extI}$. The population-wise connection strengths $J_{XY}$ characterize the recurrent connections from population $Y$ to population $X$, whereby $X, Y \in \{E, I\}$.

Previous work identified the constraints on connectivity configurations in the SSN model that underlie such nonlinear activity responses as supersaturation [15], the paradoxical effect [50, 51], bistability, and persistent activity [30]. We show that the parameters of LIF spiking networks can be mapped to the SSN such that the same activity types emerge in the spiking network, according to the observations made with the SSN. In the following sections, we discuss each activity type and its corresponding connectivity regime in the SSN, as well as in LIF spiking networks.

## Supersaturation—Firing rates can decline for growing input

Firing rates of neurons *in vivo* can show a range of nonlinear behaviors as a function of stimulus strength [52]. In particular, the activity level of sensory neurons may decrease after stimulation, and a substantial number of pyramidal V1 neurons in mice show reduced firing in response to enhanced stimulus contrast [53]. At the same time, the average activity of thalamic neurons in mice—primarily targeting V1 neurons—is an increasing function of the stimulus contrast [54]. Therefore, it appears that E neurons can be suppressed despite the increase in external input. This phenomenon is generally referred to as supersaturation [15].

First, we studied E firing response to growing inputs and aimed to delineate parameter regimes where a decreasing population response can be observed. We found that supersaturation ($\frac{d\nu_E}{d\mu_{ext}} < 0$) can be observed in a large class of connectivity and input weights configurations within the spiking networks that can be predicted by the inequality derived for the SSN model in [15, 55]

$$
r > \frac{J_{II}}{J_{EI}}.
\tag{4}
$$

Interestingly, only three network parameters determine the SSN network's ability to be in a supersaturating activity regime (Eq 4). For a network to be supersaturating, the ratio of external inputs $r$ has to exceed the ratio of recurrent inhibition $J_{II}/J_{EI}$. As a result, the remaining two

connectivity parameters (the recurrent excitation $J_{IE}$ and $J_{EE}$) cannot control the existence of supersaturating activity in the SSN. The exact point at which a network satisfying Eq 4 becomes supersaturating does, however, depend on all network parameters as it occurs when the inhibitory firing rate exceeds a specific threshold value (S1 Text, Eq S6).

To determine whether the condition derived in the SSN model (Eq S6 in S1 Text) leads to a quantitative description of supersaturation in spiking networks, we generated LIF network parameters fulfilling the supersaturating condition using the SSN-LIF mapping framework in Eq 1 (Fig 2A). We found that the activity in a LIF spiking network aligns robustly with the activity of the SSN model (Fig 2A).

Recent work [56] compared the responses of LIF and SSN models, pointing out that the peak E activity in supersaturating spiking networks is small, and in particular, it is smaller than the SSN peak. As shown in Fig 2A, the peak firing rates obtained with the two methods are in agreement. Furthermore, we show that it is possible to control the height of the E firing rate peak in both networks such that it can be made arbitrarily high (Fig 2B, S1 Text). Specifically, we show that the peak of E activity can be controlled by modifying the ratio of the external inputs $r$, and the connectivity parameters $J_{EI}$ and $J_{IE}^{-1}$ by the same factor. This manipulation derived from the SSN analysis (S1 Text) leads to the same effect in the spiking networks (Fig 2B).

To determine how close the network operates to E-I balance, we introduce the balance factor which measures the fraction of excitation that is cancelled by inhibition. We define the balance factor (BF) of population X, for positive external input $\mu_{\text{extX}}$, as $BF = \mu_{XI}/(\mu_{XE} + \mu_{\text{extX}})$. If the network operates at balance, the recurrent inhibitory input will cancel out the total excitatory input and lead to a BF of 1. It should be clarified that the balance we are considering here must be understood in the sense of *tight balance* [2, 21, 57], meaning that inhibition matches the excitation and leads to near perfect cancellation. While a partial cancellation is considered a *loose balance* [29], it does not lead to characteristic features such as a predictable linear network response. Here, the BF measured at the peak E firing rate is close to 40% for E neurons and nearly null for I neurons, demonstrating that the network operates far from E-I balance (Fig 2C). Still, the network appears to be asynchronous irregular and the firing is compatible with a Poisson spiking process as the coefficient of variation of the interspike intervals at the peak E firing rate is close to 1 ($CV_{ISI} \approx 1$) (Fig 2D). Importantly, the supersaturation regime occupies the biologically plausible activity range of 0–10 Hz in spiking networks [33–40], and the amplitude of the synaptic connection strengths, as well as the size of the network ($N = 4000$), are both in line with biological estimates of functional cortical network size [6, 58, 59] (S1 Text). We note that the supersaturation condition is incompatible with the existence of a stable balanced state solution, as defined in Eq S4 in S1 Text as it would lead to negative firing rates.

Knowing how the 2D firing rates emerge from recurrent and feedforward connectivity in the SSN allows us to invert this relation and select external inputs such that they lead to the desired E and I activity trajectory in the spiking network. This is illustrated in Fig 2E where we targeted a complex 2D trajectory. We obtained the feedforward inputs that result in the desired dynamics $v_E(t)$ and $v_I(t)$, by inverting Eq 3:

$$
\begin{aligned}
\mu_{\text{extE}}(t) &= \left(\frac{v_E(t)}{a_E}\right)^{1/n_E} + J_{EI}v_I(t) - J_{EE}v_E(t) + b_E \\
\mu_{\text{extI}}(t) &= \left(\frac{v_I(t)}{a_I}\right)^{1/n_I} + J_{II}v_I(t) - J_{IE}v_E(t) + b_I.
\end{aligned}
\tag{5}
$$

These dynamic feedforward inputs $\mu_{\text{extE}}(t)$ and $\mu_{\text{extI}}(t)$ are shown in Fig 2F, bottom and the

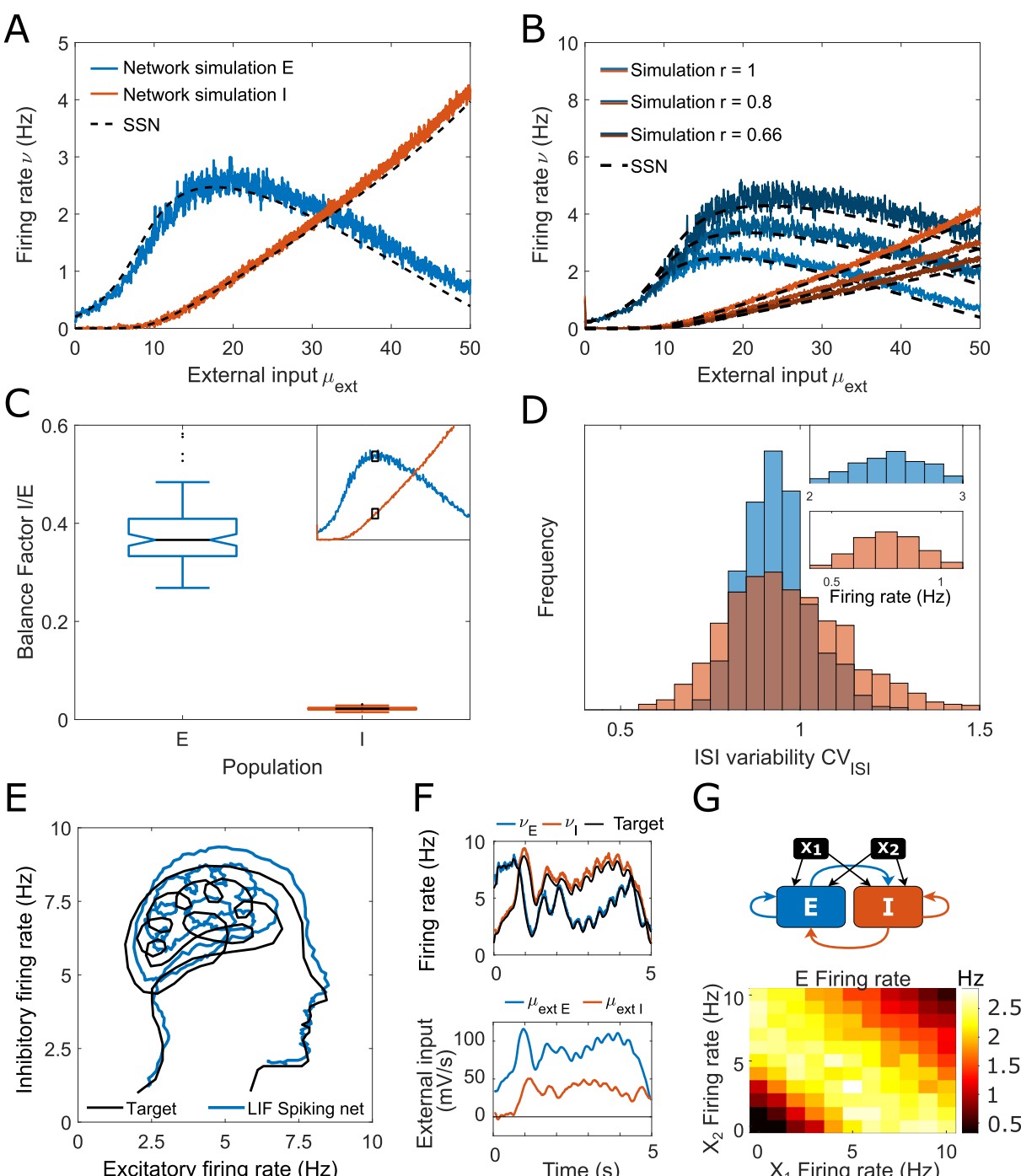

**Fig 2. SSN-predicted supersaturation can be observed in spiking network simulations.** The existence of the supersaturating response can be predicted by the SSN framework using Eq S6 in S1 Text. (A) E (blue) and I (red) firing rates as a function of external input. LIF spiking activity is in line with the SSN solution (dashed line). (B) The peak of E activity can be tuned to any desired level by modifying the I/E ratio of external inputs (*r*) along with the connectivity weights $J_{IE}$ and $J_{EI}$. (C) Boxplot of the Balance Factor (BF) for the E (blue) and I (red) neurons computed at the peak of E activity (see inset). The low values demonstrate that the network is operating far from E/I balance, as neurons in both populations receive significantly more excitation than inhibition. (D) The spiking activity of both E and I neurons is irregular and compatible with a Poisson process ($CV_{ISI}$ close to 1). The inset shows the distribution of firing rates of individual neurons. (E) Spiking networks can follow a user-defined target dynamical trajectory. The black line shows the target trajectory we aim to replicate with the network. The blue line shows the trajectory of the spiking network in the E-I activity phase space. (F) Same simulation as in panel E. The time course of the E and I firing rates in the LIF network (top) follows the target trajectory and results from designed dynamical inputs (bottom). (G) Supersaturating networks can perform the XOR task. Top: Layout of the network used to perform the XOR task, where the E-I network is supersaturating (Eq S6 in S1 Text). Bottom: The LIF E population activity performs XOR logical operation of the two inputs $X_1$ and $X_2$. The feedforward weights are $J_{EX_1} = J_{IX_1} = J_{EX_2} = J_{IX_2} = 2.5$ mV. The spiking network parameters can be found in Table B in S1 Text.

fidelity of the targeting is illustrated in Fig 2F, top. Notably, the timescale of the autocorrelation function of neuronal activity (as defined in [60]) is around 300 ms, which is in line with recorded cortical activity [47–49]. These results indicate that complex dynamic trajectories evolving on biologically realistic timescales can be accurately captured by the SSN steady states Eq 3. It follows that the mapping between the steady states of the SSN and spiking neural networks provides a valuable approximation even for slow spiking network dynamics.

Let us note that while we used here dynamic feedforward inputs to move along the activity trajectory, it is equally possible to dynamically modify the connectivity to obtain the same 2D trajectory in activity space. In this scenario, synaptic plasticity is recruited to obtain a user-defined output. This can be done by setting the plastic connections $J$ as dynamic while the external inputs are constant.

$$
\begin{aligned}
J_{EE}(t) &= \frac{\left(\frac{v_E(t)}{a_E}\right)^{\frac{1}{n_E}} + J_{EI}v_I(t) - \mu_{ext} + b_E}{v_E(t)} \\
J_{IE}(t) &= \frac{\left(\frac{v_I(t)}{a_I}\right)^{\frac{1}{n_I}} + J_{II}v_I(t) - r\mu_{ext} + b_I}{v_E(t)}.
\end{aligned}
\tag{6}
$$

Overall, we show that the mapping between SSN and spiking networks makes it possible to construct inputs or synaptic weights in a spiking neural circuit such that its activity follows a user-defined complex target dynamical trajectory.

In balanced networks, the implementation of logical gates is a complex task due to the linearity of the transfer function [61]. Therefore, we asked whether the nonlinear regimes of spiking networks can be used to perform specific logical operations. Here, we show that it is possible to combine feedforward and recurrent inputs in a way that makes the circuits perform the nonlinear XOR operation, which is one of the key computing components of logical circuits, while being challenging to implement in a neural network [62]. We show in Fig 2G how a supersaturating network can execute the XOR operation from two input signals. The E activity is maximal if the input $X_1 + X_2$ corresponds to the peak input in the SSN supersaturating regime. The E activity is unstimulated if both inputs $X_1$ and $X_2$ are low and silenced if they are both high. This shows that the nonlinearity of biologically-sized spiking networks can be exploited to carry out fundamental logical operations.

We have shown that the SSN provides a powerful framework to study the supersaturating network presented in Fig 2. Next, we explore if a neural network corresponding to the experimentally reported connectivity parameters in mouse V1 by the Allen Institute [6] can also show supersaturation (Table B in S1 Text). We use these parameters as a reference point for the biologically plausible range of connection strengths. Interestingly, a circuit with these connectivity parameters does not have a balanced state solution for the equal external input ratio $r = 1$ ($\mu_{extE} = \mu_{extI}$) and requires $r < 0.9$ to fulfill the balanced state requirement (Eq S2 in S1 Text). The network can be supersaturating for values of $r$ larger than 0.9. Remarkably, for $r = 1$, the E activity does not decrease in the low input range but saturates instead (Fig 3A), the network only becomes supersaturating for $\mu_{ext} > 150$ mV/s (S1(B) Fig). For larger values of $r$ the activity decreases and becomes silent for inputs close to 100 mV/s (see inset Fig 3A, $r = 1.5$).

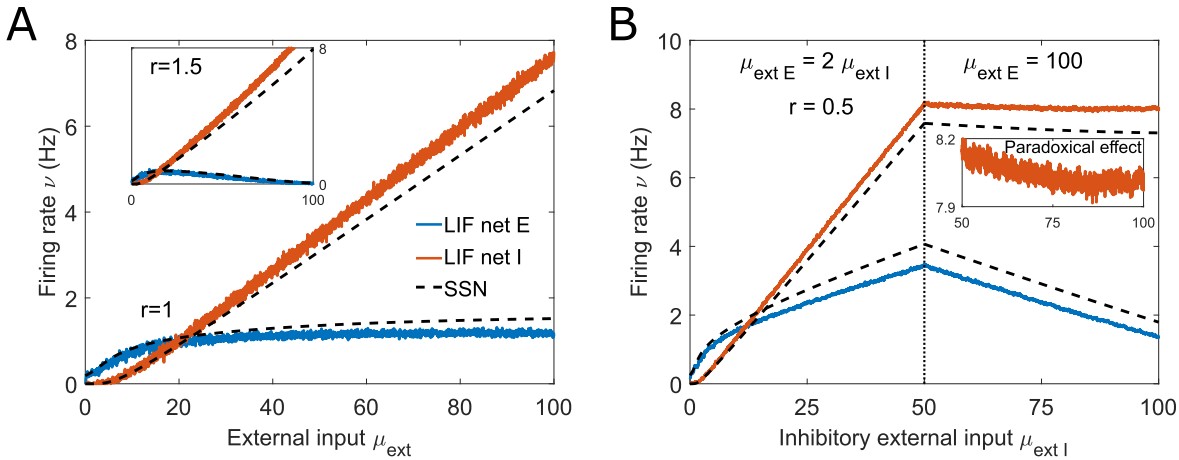

**Fig 3. The experimentally reported network parameters can generate supersaturation and be adapted to enter the inhibition-stabilized state.** (A) Firing rates of the E and I populations, blue and red lines, respectively, as a function of external input using the parameters reported by the Allen institute [6], see Table B in S1 Text. The dashed lines show the SSN solution. In this network, the E activity saturates for inputs larger than 20 mV/s. If the external input to the I population is larger than that to the E population, as shown in the inset with $r = 1.5$, E firing rate declines for growing input. (B) This spiking network exhibits inhibitory stabilization matching the predictions of the SSN (dashed line). The connection strength $J_{EE}$ is higher than in panel A (Table B in S1 Text). The ISN state is exposed by the paradoxical effect which occurs when the I firing rate decreases for increasing $\mu_{extI}$. First, the inputs to both populations grow to drive the network in a state where the E subnetwork is unstable: $\nu_E > 1.5$ Hz, Eq 7 (vertical dotted line). Once $\mu_{extI}$ reaches 50 mV/s, only the input to I increases (from 50 to 100 mV/s), while $\mu_{extE}$ remains at 100 mV/s. This results in a decrease in the firing rates of both populations, as predicted by the SSN (dashed lines). The inset shows a close up of the I activity to illustrate the paradoxical effect and shows that the paradoxical effect wanes as the E firing rate approaches the ISN threshold ($\nu_E \approx 1.5$ Hz).

## Inhibitory stabilization and its presence for reconstructed synaptic weights

Inhibitory stabilization is a network state in which the recurrent excitation feedback loop is strong and intrinsically unstable but can be stabilized by the recurrent inhibition [31, 63]. The paradoxical effect is a feature of the ISN [31], in which the I activity decreases as the input to the I population is increased ($\frac{d\nu_I}{d\mu_{extI}} < 0$). Recent studies using optogenetic stimulation of inhibitory neurons confirmed the paradoxical effect in mouse visual, somatosensory, and motor cortices [64] suggesting that the ISN is a ubiquitous property present across cortical networks. A recent review presented further experimental evidence and techniques used to study the inhibition-stabilized dynamics and discussed the ISN consequences for cortical computation [32]. In the SSN model [50, 51], a network is inhibition-stabilized if it fulfills the condition

$$\nu_E > \left(a_E n_E^{n_E} J_{EE}^{n_E}\right)^{-\frac{1}{n_E - 1}}. \tag{7}$$

We note that in networks with a threshold linear transfer function, the analogous ISN condition only requires a strong recurrent coupling $J_{EE} > 1$ and does not impose any constraints on the E firing rate level or the transfer function parameters [31, 32]. However, large enough $J_{EE}$ does not always guarantee that a recurrent neural network with a nonlinear transfer function is in the ISN regime. Increasing $J_{EE}$ can also lead to instability, as the excitatory feedback loop can strengthen to a point where it escapes stabilization from recurrent inhibition. In the extreme case, it is even possible to build a network that can never enter the ISN regime regardless of the value of $J_{EE}$, as E activity never reaches the level where it can be stabilized by inhibition (S1 Text).

Next, we investigated whether this condition (Eq 7) can predict the existence of the ISN in spiking networks of LIF neurons. Interestingly, we found that the ISN condition cannot be

met for the connectivity strengths reported for mouse V1 from the Allen Atlas [6] if the E/I input ratio $r$ is equal to 1. This is due to the fact that the required E firing rate ($\nu_E$ >27 Hz) is higher than the maximum stable E firing rate reached in the network (Fig 3A). Yet, even for very low values of $r$ (around $r$ = 0.1), an ISN state can only be reached by exposing the network to very high external inputs (around $\mu_{ext}$ = 1000 mV/s) (S1(B) Fig). We will choose a network which operates outside these cases since the corresponding firing rates are far beyond the 0–10 Hz firing rate range we consider in this study. Therefore, in order to illustrate the ISN condition, we modified one of the connectivity strengths. Specifically, we increased the connectivity parameter $J_{EE}$, which is supported by the study by [10] who report larger $J_{EE}$ than the Allen Atlas [6]. We set $J_{EE}$ such that the network is in the ISN state for E firing rates larger than 1.5 Hz and kept all other connectivity strengths as reported by the Allen Atlas ([6], see Table B in S1 Text). Fig 3B shows that the resulting network exhibits the paradoxical effect and is therefore in the ISN regime.

## Bistability and persistent activity

One of the most prominent experimentally recorded neural activity features *in vivo* is the network ability to switch between higher and lower firing levels. One example is spontaneously alternating intervals of tonic firing and silence observed across different cortical areas [37]. Another example is the sustained firing rate in the prefrontal cortex after stimulus withdrawal during decision-making tasks which is hypothesized to represent short-term memory [14, 65]. The coexistence of multiple network states can be explained theoretically by bistability, where the system has two stable states for the same level of input. If multiple stable states co-exist in a network model, a sufficiently large perturbation can drive network activity away from its current state towards another attractor. In the situation where a bistable network can sustain its high activity level in the absence of feedforward input, it has persistent activity. Here, we asked whether the SSN model can predict the connectivity regime supporting bistability in spiking circuits.

Bistability and persistent activity can be obtained in the SSN model [30] without the need for synaptic plasticity [66] or complex synaptic weight distributions [67]. Unlike supersaturation and inhibition stabilization, bistability cannot be delimited by a simple tractable condition on network parameters (S1 Text). However, we can use the conditions presented in [30], as a starting point to guide our search for bistability in biologically realistic spiking networks, even though they are derived under restrictive assumptions on the $a$, $b$, and $n$ parameters. We show an example of a biologically realistic bistable network in Fig 4.

The LIF network simulation confirms the SSN-predicted bistability Fig 4: the network can sustain either low activity or high activity for external inputs in the 2–4 mV/s range. Although the SSN rate model is deterministic, the spiking network simulation is not. Due to the stochastic nature of the neuronal activity, fluctuations in firing can cause spontaneous transitions between steady states (shown in Fig 4A, inset). We note that the spontaneous transitions between the up and down states have not been reported in the bistable balanced networks with short-term plasticity [66]. This is because the fluctuations driving spontaneous transitions are finite-size effects [20], and the switching probability decreases with network size.

We find that a higher excitatory membrane time constant broadens the window of bistability (Fig 4B), making bistability more robust to spontaneous fluctuations and easier to locate in phase space. As $\tau_E$ increases, the bistability window shifts towards lower feedforward input. When the bistability window intersects the vertical $\mu_{ext}$ = 0 axis, the network has a persistent activity state in the absence of feedforward input ($\tau_E$ = 22 ms in Fig 4B). Here again, the E-I

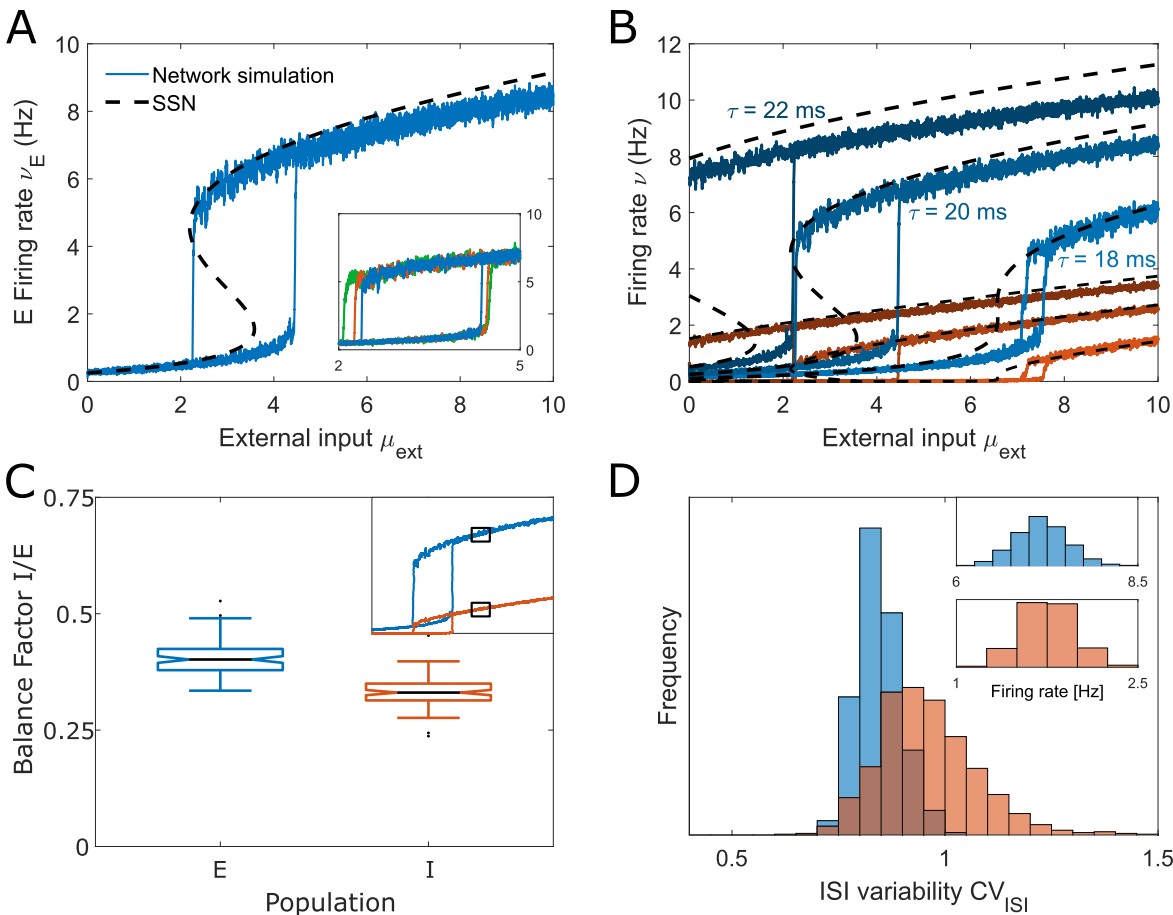

**Fig 4. SSN-predicted bistability and persistent activity can be observed in spiking network simulations.** (A) E and I firing rates as a function of external input in a bistable network (coexistence of high and low activity states for a given external input). Simulated LIF spiking activity is accurately predicted by the SSN. The inset illustrates that the width of the bistability window can vary between simulations of the same network due to the spontaneous transitions between the two states. (B) The width of the bistability window depends on the excitatory membrane time constant: higher values of $\tau_E$ lead to broader bistability windows, which are shifted leftward. If $\tau_E$ is sufficiently large for the bistability window to exist for zero feedforward input, the network can sustain persistent activity. In the SSN, changes in $\tau_E$ correspond to changes in the parameters $a_E$, $b_E$ and $n_E$. (C) Balance Factor for the E (blue) and I (red) neurons measured on the high activity branch (see inset). The BF values are far from 1, which indicates that the network is only loosely E/I balanced. (D) The coefficient of variation of the interspike intervals ($CV_{ISI}$) is near 1, which is compatible with a Poisson process and demonstrates that activity is asynchronous and irregular. Both (C) and (D) are measured in the upstate, as shown in the inset to (C). All parameters are given in Table B in S1 Text.

balance is weak, as shown by the balance factor of inputs (Fig 4C), and the spiking activity is Poisson-like, as shown by the coefficient of variation of the ISIs Fig 4D.

Finally, the nonlinear transformation performed by spiking networks can be functionally relevant for information processing. Logical operations such as the AND operation can be implemented without the need to recruit synaptic plasticity, thanks to the sharp transition between the two stable states. If the transition from the low to the high activity level requires a strong input, so that two signals $X_1$ and $X_2$ need to be present to elicit the transition, the network can execute the AND operation. Moreover, the bistability of a neural network can also offer the possibility to store information. Once the network has been switched into a different activity state by a strong perturbation, it remains in the same state even after the perturbation withdrawal.

## Computational regimes and their position in input space

In previous sections, we demonstrated that the SSN framework can be used to locate specific computational regimes such as supersaturation and the paradoxical effect in parameter space, and that the observations derived from the SSN are confirmed in corresponding spiking neural networks. Here, we focused on the activity regimes associated with the 2D space of the feedforward input and the I/E external input ratio ($\mu_{\text{ext}}$, $r$). For two examples of connectivity matrices $J$, we scanned the 2D input space for supersaturation (Eq S6 in S1 Text), ISN (Eq 7) and bistability using the characteristic function $\mathcal{F}$ as defined in [30] (Eq S10 in S1 Text). We also show the input regimes for which the network permits a balanced limit solution (Eq S2 in S1 Text). Importantly, the sign of the determinant of the weight matrix (det $J = J_{EI}J_{IE} - J_{EE}J_{II}$) determines the number of SSN steady states (stable and unstable): for det $J > 0$, the SSN has an odd number of steady states whereas for det $J < 0$ the number of steady states is even [15, 30]. Thus, networks with positive det $J$ (Fig 5A) always have at least one steady state. In the network shown in Fig 4, bistability occurs when the system transitions from having one steady state to having three (two stable and one unstable). On the other hand, networks which have a negative det $J$ (Fig 5B) can lack steady states at all, and any possible stable steady state coexists with an unstable steady state [30]. Finally, the sign of det $J$ also determines whether the system can have a stable balanced state (Eq S4 in S1 Text) or lacks it. In networks where the sign of det $J$ is negative, a balanced state solution can exist with positive firing rates if $r > \max\left(\frac{J_{II}}{J_{EI}}, \frac{J_{IE}}{J_{EE}}\right)$, but it is unstable [28].

Fig 5 shows the map of feedforward inputs and the corresponding computational regime for two examples of the connectivity matrix $J$. Panel A corresponds to the connectivity parameters from the bistable network shown in Fig 4 with det $J > 0$. The region where bistability is expected corresponds to the results in Fig 4A with $r = 1$. The balanced state exists and is stable for low values of $r$. Panel B corresponds to a network with det $J < 0$. In this case, the balanced state only exists for high values of $r$, but it is unstable. Furthermore, we find that for large input

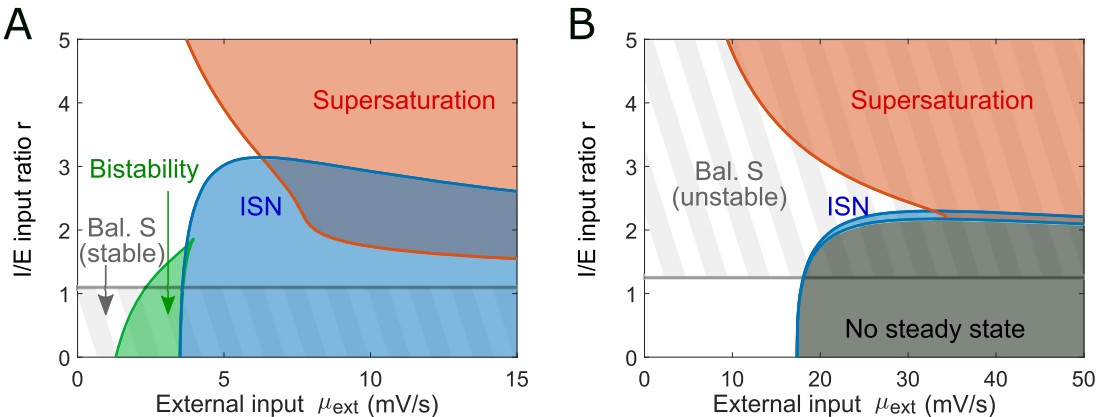

**Fig 5. Mapping the computational states in the SSN model for two representative connectivity regimes.** (A) Varying the ratio of the external input weights and the amplitude of external drive in a network with a positive det $J$ allows to traverse different computational regimes ($J$ as in Fig 4, see parameters in Table B in S1 Text). Gray stripes denote the input space subset with a stable balanced state limit ($N \rightarrow \infty$) which does not exist above the horizontal gray line. The blue area represents the inhibition stabilized regime (ISN). The red area denotes the phase space occupied by supersaturating spiking activity. The green area corresponds to a bistable region (as shown in Fig 4A with $r = 1$). Within the green region, the up-state is in the ISN whereas the down-state is not. We note that the inhibitory stabilization and supersaturation can co-exist. (B) The same analysis is performed on a network with negative det $J$ (Table B in S1 Text). In this case the balanced state limit is unstable. The ISN region is narrower and there is a broad range of inputs for which the network does not have a steady state solution.

and low $r$, the network does not have a steady state. In this region, inhibition cannot stabilize the network, and the activity blows up. The same analysis is also performed for the supersaturating network in Fig 2 and the mouse V1 network in Fig 3A (S1 Fig). Overall, using the SSN model, we can precisely locate the regions corresponding to distinct behaviors of spiking networks in their parameter space. Notably, we observe that the sign of the determinant of the connectivity matrix $J$ plays a crucial role in the type of activity regimes available to the network (S1 Text).

## Effect of network size on network response nonlinearity

While biologically-sized networks can generate diverse nonlinear responses to external input, the balanced state framework implies that network response becomes linear as network size approaches infinity. How do networks transition from nonlinear to linear regimes for increasing network size $N$? To tackle this question, we re-scaled the recurrent connections $j_{XY}$ by the factor $1/\sqrt{N}$ as a function of network size $N$, and increased $N$ from $N = 4 \times 10^3$ to $5 \times 10^5$ while keeping the connection probabilities fixed, leading to an effective re-scaling of the population-wise $J_{XY}$ by $\sqrt{N}$. This parameter re-scaling follows the convention of the balanced state theory [2, 21] and allows us to address whether these nonlinear spiking networks converge to the expected balanced state, and if so, when and how. In the balanced state convention, the feedforward input follows the same rescaling and grows with $\sqrt{N}$. We show in Fig 6A, 6C and 6E the network response to the effective feedforward input after scaling ($\mu_{ext}$), and in Fig 6B, 6D and 6F the network response as a function of the external input before scaling, ($\mu_{ext}/\sqrt{N}$) to highlight a possible convergence to the balanced solution as $N$ grows.

A dynamically stable balanced state limit can only exist if det $J$ is positive and the fraction of external input weights $r$ satisfies $0 < r < \min\left(\frac{J_{II}}{J_{EI}}, \frac{J_{IE}}{J_{EE}}\right)$, see Eq S4 in S1 Text. In our network convergence study, we focus on three spiking networks: one supersaturating network with det $J > 0$ (shown in Fig 2), one bistable network with det $J > 0$ (shown in Fig 4), and a supersaturating network with det $J < 0$ (presented in Fig 5B, where we set $r = 3$). Among our three example networks, we have one example for which a balanced state limit does not exist (supersaturation), one network with a stable balanced state solution (bistable network) and one network with a balanced state that exists but is dynamically unstable (det $J < 0$). In all three cases, the SSN model remains an accurate description of the spiking network mean activity across different network sizes $N$ (Fig 6A, 6C and 6E), and its predictions align qualitatively and quantitatively with the self-consistency solution $\Phi_{sc}$ model (Fig 6B, 6D and 6F, inset).

We find that the network response can remain nonlinear even for very large network sizes consisting of up to half a million neurons (see inset). By considering the feedforward input before scaling ($\mu_{ext}/\sqrt{N}$) [2, 21], the network response should converge toward a single linear solution—the balanced limit (Eq S1 in S1 Text, dashed line in Fig 6D and 6E). In the case of the supersaturating network, the balanced limit does not exist, as it would lead to negative E firing rates. Therefore, in the limit of large network size the E firing rate tends to zero. In the case of the bistable network, a balanced state limit does exist but the network response is still far from converging to it, even for $N = 5 \times 10^5$. Finally, for the network with det $J < 0$ and $r = 3$, the network exhibits supersaturation for $N = 4000$ (see Fig 5B). However, as $N$ increases, the network enters a region for which there is no steady state, and where the firing rates blow up. This behaviour is observed in spiking network simulations, the SSN solution and the mean field $\Phi_{sc}$. The inset shows how this instability is caused by the collision of the two steady state branches, leaving a gap where the firing rates are unbound. For this network, the mean-field solution converges to the balanced limit as $N$ increases (Fig 6F). However, the balanced state

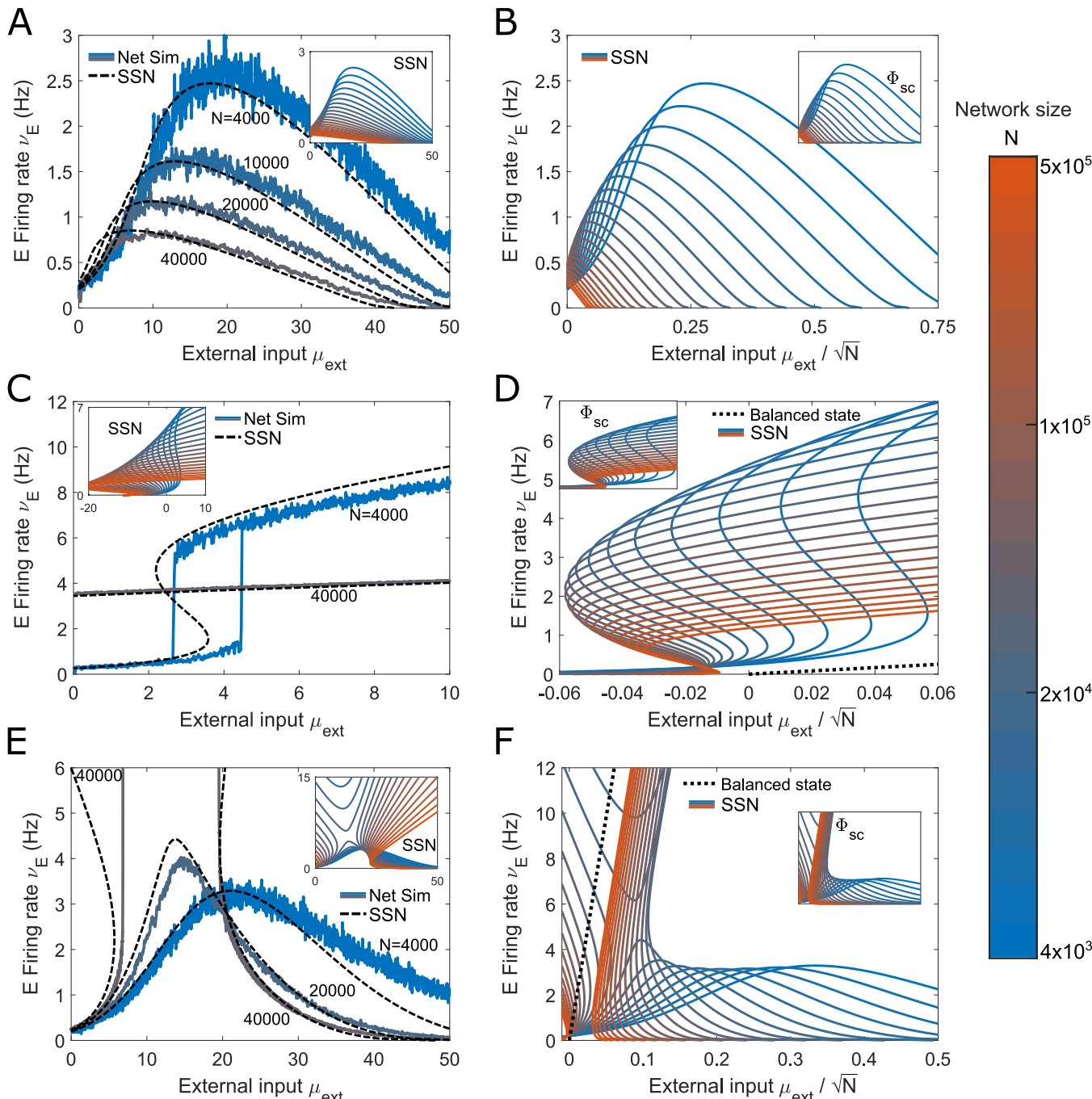

**Fig 6. Increasing network size does not guarantee convergence to a balanced state.** (A-B) Supersaturating network with det $J > 0$, (C-D) bistable network with det $J > 0$, (E-F) supersaturating network with det $J < 0$, parameter regimes which we identified using the SSN framework and studied for $N = 4000$ in previous figures. Here, we gradually increase the size of these networks $N$ and follow the balanced network convention to rescale the weights $J_{XY}$ by $\sqrt{N}$ as the network grows. (A) The excitatory firing in the supersaturating network from Fig 2A across different network sizes. The colored lines represent spiking networks (from $4 \times 10^3$ to $4 \times 10^4$), the black lines represent the corresponding SSN solution. The inset shows the excitatory rate for network sizes from $N = 4 \times 10^3$ (blue) to $N = 5 \times 10^5$ (red), in steps of $\times 10^{0.1}$ obtained using the SSN. (B) The same network as in A, but the external input is expressed before scaling: $\mu_{ext}/\sqrt{N}$. This network does not have a balanced state solution (see first condition of Eq S4 in S1 Text). As $N$ grows, the excitatory activity peak becomes smaller and in the limit of very large networks, the excitatory population remains silenced ($\nu_E = 0$). The inset shows that the Ricciardi self-consistency solution $\Phi_{sc}$ (Eq 14) and the SSN model predict the same behavior as $N$ increases. (C) The excitatory activity for the bistable network from Fig 4 for $N = 4 \times 10^3$ and $N = 4 \times 10^4$. The spiking activity of the spiking LIF network (colored lines) is captured by the SSN model (black lines). As $N$ increases we observe a broadening of the bistability window and a decreasing firing rate, see inset. (D) The same network as in panel C but now without external inputs scaling to highlight the balanced solution. The balanced state predicts a linear solution for the $N \to \infty$ limit (Eq S1 in S1 Text, dashed line). The convergence to the

balanced limit is very slow, and even at $N = 5 \times 10^5$ neurons, the network rates still do not converge to the balanced state. (E) Activity of a spiking network with a negative det *J* from Fig 5B, with *r* = 3. As *N* grows, the excitatory activity dissociates into two distinct branches separated by an unstable region where the firing rates diverge to $\infty$. The inset illustrates that this separation occurs when the stable and unstable steady states collide. (F) The same network as in E, now with the external inputs expressed before scaling. The balanced solution of this network (dashed line) is unstable (Eq S4 in S1 Text). The unstable network solution approaches the balanced state, while the stable solution tends to 0 as *N* increases.

limit is unstable here, and it only matches with the unstable mean-field solution (high firing rates part of the branch) whereas the stable low activity solution tends to zero.

Overall, our example networks illustrate that for many classes of spiking networks with biologically plausible sizes and connectivity configurations, the activity will escape the predictions of the balanced state. Depending on the parameters, the balanced state may not exist or be unstable. Yet, even when a stable balanced state exists, it is not guaranteed that it provides a realistic description of network activity, even for unrealistically large sizes *N*. Next, we investigated whether these networks which are non-linear and do not conform to the balanced state prediction can have a balance factor close to 1. We have shown that the values of BF are far from 1 in the bistable network of 4000 neurons (Fig 4C), which is an indicator that the network is operating far from balance. However, we found that the BF of inputs can be network size dependent: increasing the size *N* from 4000 to 40000 strongly increased the inhibition to excitation ratio even though the firing rate activity in both cases does not conform to the balanced state solution (S2 Fig). In summary, the observation of significant cancellation of incoming excitatory and inhibitory signals does not guarantee that the balanced state framework is applicable to predict the firing activity. Even small deviations from tight balance ($BF <1$) can lead to significant deviation in the resulting network activity regime.

## Discussion

Understanding how activity regimes of biologically-sized spiking networks relate to network structure is critical to making sense of experimentally recorded data. The state-of-the-art experimental techniques now enable simultaneous recordings of thousands of neurons [68, 69]. These large experimental datasets require solid theoretical foundations bridging knowledge on the spiking network composition with the observed network activity. Here, we show how to predict the computational regime of a spiking network comprising a few thousand neurons from its connectivity configuration by mapping the spiking network to a tractable SSN model. This network size corresponds to a fundamental functional network unit such as a minicolumn found in diverse cortical regions [70–74]. Additionally, we set the range of firing rate activity to meet the experimentally reported range of a few Hz [33–40].

In the present work, we developed a mapping between the rate SSN model and a biologically-sized spiking network of two neuronal populations without any constraints on the network's connectivity configurations. We have shown that the nonlinear behavior of the spiking networks can be mechanistically understood using the lower-dimensional and mathematically tractable SSN model. The rich computational repertoire of the SSN model originates from one simple experimentally inspired assumption that the activation function of individual neurons is supralinear. This function resembles the input-output relation in a spiking network, providing a critical component for mapping the two models. Using the mapping, we delineated connectivity regimes for which nonlinear computations such as bistability, supersaturation, inhibitory stabilization, or even the absence of steady states can occur in spiking networks. We found that networks can be inhibition stabilized in conditions where a balanced limit does not exist, even though both require strong inhibitory feedback. For example, the inhibitory stabilization can overlap with the supersaturation regime and is

possible in networks with the negative determinant of the connectivity matrix (det $J < 0$, Fig 5), a connectivity regime for which a balanced network solution would be unstable. We found that a set of connectivity configurations obtained from experimental connectivity estimates fulfills the balanced state condition only if the E population receives a stronger feedforward input than the I population. Additionally, we have shown that a spiking network could implement an XOR gate in a supersaturation connectivity regime which we detected using the proposed mapping between SSN and spiking network models (Fig 2G). This observation provides a mechanistic understanding of how a trained spiking network could implement a fundamental XOR logical gate.

Overall, we found that the SSN model can accurately predict nonlinear activity regimes of biologically-sized spiking networks without constraints on the network connectivity configurations imposed by the balanced state condition. Specifically, the SSN model can describe the mean response of biologically-sized spiking networks that are not large enough to converge to the balanced state limit or cannot reach the limit because their underlying connectivity does not fulfill the balanced state condition (Fig 6). Since the SSN model has fewer dimensions than spiking network models, we expect that spiking network simulations may exhibit activity regimes which cannot be obtained in the SSN model. However, we found that the provided mapping can accurately approximate the mean activity of a spiking network for a set of nonlinear computational regimes supported by the SSN. A recent study [56] pointed to discrepancies between the SSN and spiking network outcomes in the supersaturation regime. Specifically, numerical simulations in [56] suggested that the supersaturation peak in the SSN model's output seems generally smaller than the peak reached by a spiking network. Here, using our SSN to spiking network mapping formalism, we show how to align and simultaneously control the supersaturation peak of both the SSN and spiking network models Fig 2B.

In this work, we choose the SSN rate model to provide a mathematically tractable description of spiking neural networks and uncover diverse nonlinear activity regimes. It should be noted that alternative models derived from the balanced state could be used as well. For example, balanced networks with short-term synaptic plasticity have been proposed to permit the emergence of nonlinear activity, such as bistability [66]. Likewise, the experimentally reported small feedforward input which drives spiking activity *in vivo* [75–78] was inconsistent with the original balanced state predictions but was accommodated via the inclusion of broad synaptic weight distributions [67]. Similarly, semi-balanced networks were proposed [61], where neurons which receive net inhibition remain silent. This generates a piecewise-linear manifold which can operate as a nonlinear decision boundary and allows for a broader domain of validity than the classical balanced framework. While the network nonlinearities we report in this work only emerge as a consequence of powerlaw-like transformation of the LIF neuron, future works could aim to include the effects of synaptic plasticity, broad distributions of synaptic weights or silencing of neuronal subpopulations in a tractable mathematical framework to study the interplay of these multiple sources of nonlinearity.

Similarly, a recent article by Sanzeni et al. [56] used a different approach to study nonlinearities in spiking neural networks. The self-consistency solution was analyzed by using expansions of the Ricciardi $\Phi$ function in two limits where nonlinearities in the F-I curve occur: at response-onset, where firing rates are low and spiking is driven by noise, and at saturation where firing rates reach a maximum bounded by the refractory period of neurons. These two nonlinearities can lead to supersaturation and multisolution at the network level. While we do not consider saturation and refractory periods, our approach expands on these results by providing an alternative framework to describe the noise-induced nonlinearity through the power-law approximation. This power-law has a broader range of validity since it does not

require to operate at the weak coupling limit. Furthermore, the power-law function is designed to be both an accurate approximation of LIF dynamics and a simple mathematical expression. Thanks to this property, we showed that the SSN model can accurately match the activity of biologically realistic networks of LIF neurons including possible nonlinear features with tractable equations and few parameters. This framework is useful to develop theoretical results which provide a deeper understanding of network mechanisms than numerical simulations. Nevertheless, it should be noted that the SSN, being a rate model, does not account for the recurrent noise originating from the activity in the network, which has been shown to make or break some nonlinear regimes [56]. Overall, the two approaches are complementary as the power-law framework focuses on mathematical tractability, matching LIF simulations and uses a unique activation function over the biological activity range while Sanzeni et al. [56] provide a finer analysis at two precise sources of nonlinearity.

Concurrently, large-scale computational projects have developed detailed numerical simulations by including the state-of-the-art activity datasets and connectivity reconstructions to precisely recreate a mammalian nervous system [79]. Due to the complex biology of the brain, the resulting network simulations represent multidimensional dynamical systems whose behavior often cannot be predicted and controlled. In contrast, our approach of mapping a multidimensional spiking network to a lower-dimensional mathematically tractable circuit provides promising access to a mechanistic understanding of complex dynamical systems such as the mammalian brain. Future studies could expand the results presented here by including additional neuronal populations into the network, considering the heterogeneity of neuronal cell types or of connection strengths affecting the network dynamics [67, 80] or by including plasticity in synaptic connections. Similarly, future work could analyze the dynamical properties of spiking networks ($\tau_{P_E}$ and $\tau_{P_I}$ in Eq 2) to study such activity regimes as oscillations observed in the SSN model [30] and characterize the dynamical stability of network states or expand the model to characterize finite-size effects which become substantial in smaller networks [17, 19, 20]. Finally, the study of biologically-sized spiking networks ($\sim 10^3$ neurons) provides an understanding and control of functional units such as minicolumns or layers which compose larger networks corresponding to whole functional brain areas and beyond ($\sim 10^5$ neurons [81]). Progress in experimental techniques requires computational models clearly explaining the relationship between activity and connectivity datasets. Here, the necessity for a better understanding of the brain's fundamental building blocks—such biologically-sized spiking networks—remains a critical milestone in exploring the brain's global functions and working principles.

## Materials and methods

### Power-law approximation of the input-output transformation in a single neuron

We represent the spiking activity of a neuron using the integrate-and-fire model

$$\frac{dV}{dt} = -V/\tau + I. \tag{8}$$

Where $V$ is the membrane potential, $\tau$ is the membrane time constant, and $I$ is the input to the neuron. Upon reaching the firing threshold $\Theta$, $V(t)$ is reset to $V_R$. In the following, we assume that the input $I$ received by the LIF neuron is white noise, which can be written as

$$I(t) = \mu + \sigma\eta(t).$$

Where $\mu$ is the mean input, $\sigma$ is the noise strength and $\eta$ is a normally distributed random

variable, such that

$$\langle \eta(t) \rangle = 0$$
$$\langle \eta(t)\eta(t') \rangle = \delta(t - t').$$

Under the assumption of white noise, the firing rate of the neuron in Eq 8 can be described by the Ricciardi transfer function $\Phi$ [42, 82]

$$v = \Phi(\mu, \sigma, \tau) = \left( \tau\sqrt{\pi} \int_{\frac{V_R - \mu\tau}{\sigma\sqrt{\tau}}}^{\frac{\Theta - \mu\tau}{\sigma\sqrt{\tau}}} e^{z^2}(1 + erf(z))dz \right)^{-1}. \tag{9}$$

For low inputs, $\Phi$ is a supralinear function of the mean input $\mu$ and can be accurately approximated by a power law with an exponent $n > 1$ (see Fig 1). For high inputs, however, $\Phi$ becomes linear. In this work, we restrict ourselves to the low firing rate regime ($v \leq 10$ Hz) often reported for cortical activity measured *in vivo* [33–40]. In this low activity regime with constant noise $\sigma$ and time constant $\tau$, the firing rate can be accurately approximated by a power law Eq 1, see Fig 1B. The parameters $a$, $b$ and $n$ are obtained by fitting $\Phi$ Eq 9 (Fig 1C) with a power law.

## LIF spiking network simulation

We consider a spiking network of one excitatory (E) and one inhibitory (I) population with $N_E = \frac{3}{4}N$ and $N_I = \frac{1}{4}N$ LIF neurons, respectively. We assume that both E and I populations are homogeneous, i.e. neurons within each population have the same parameters (membrane time constant $\tau_X$, threshold potential $\Theta_X$, reset value $V_{RX}$), receive external input with the same mean $\mu_{\text{extX}}$ and noise $\sigma_{\text{extX}}$, $X \in \{E, I\}$. The E and I populations have different membrane time constants (see black crosses in Fig 1C), and the feedforward input they receive differs by a factor of $r$ ($\mu_{\text{extI}} = r\mu_{\text{extE}}$, see Fig 1A). Additionally to the feedforward input, the neurons receive recurrent input from other E and I neurons in the network. The connections are randomly generated based on a homogeneous probability of connection, such that each neuron in population $X$ receives inputs from exactly $N_Y p_{XY}$ randomly chosen neurons in population $Y$, where $p_{XY}$ is the connection probability from population $Y$ to population $X$. We use two types of synapses, the delta synapse and the exponential synapse.

For delta-synapses, the function

$$I_{XY}(t) = j_{XY}\delta(t - t_s) \tag{10}$$

represents the input from a neuron of the population $Y$ to a neuron in $X$. Where $j_{XY}$ is the strength of the synapse, $t_s$ is the spike time of the presynaptic neuron, and $\delta$ is the Dirac delta function.

In some network configuration, delta synapses promote synchronization of the whole neuronal population. This synchronicity can lead to population spikes [16, 83] which violates the assumption of asynchrony and irregularity in the mean field approach. In order to avoid this synchronization in these cases, we use exponential synapses instead of delta synapses. In exponential synapses, the synaptic potential from a neuron in population $Y$ to a neuron in population $X$ decays exponentially in time

$$I_{XY}(t) = \frac{j_{XY}}{\tau_{sXY}}e^{-\frac{t - (t_s + D)}{\tau_{sXY}}}, \quad t > t_s + D. \tag{11}$$

Where $j_{XY}$ is the strength of the synapse, $t_s$ is the spike time of the presynaptic neuron, $\tau_{sXY}$ is the synaptic decay time constant and $D$ is the synaptic delay. This type of synapse prevents synchronization as the effect of each spike is more distributed in time and each synaptic connection has a different delay $D$.

We use the exponential synapse in the spiking network simulation in Figs 4, 6C and 6E and S2 Fig, and delta synapses in all other cases. The exponential synapse parameters can be found in Table C in S1 Text. It should be noted that the synaptic delay values we use are higher than experimentally reported [84–86], but this jitter only serves to prevent population synchrony and does not affect the steady state network activity.

## Self-consistent network solutions

In this work we derive predictions for the activity regimes of spiking networks using the closed-form solutions offered by the SSN framework (Eq 3). In some instances, it is useful to compare the SSN predictions to the previously proposed self-consistent network solutions to understand the dynamic origin of the SSN predictions. The self-consistent system is derived from the single-neuron LIF response. For a neuron embedded in a network, the input it receives ($I$ in Eq 8) is the sum of feedforward input ($I_{ext}$) originating from outside of the network and recurrent input ($I_{rec}$) caused by synaptic connections from other neurons in the considered network. Here we assume that feedforward input is white noise with mean $\mu_{ext}$ and noise strength $\sigma_{ext}$. The recurrent input results from the spike trains of presynaptic neurons.

$$
\begin{aligned}
I &= I_{rec} + I_{ext} \\
I_{ext} &= \mu_{ext} + \sigma_{ext}\eta(t) \\
I_{rec} &= \sum_{syn} j_{syn} \sum_{spike} \delta(t - t_{spike}).
\end{aligned}
$$

Where the first sum considers all synapses onto a neuron and the second sum considers all the spikes arriving at this synapse. For Poisson spike trains, the mean and variance of $I_{rec}$ for a neuron in population X can be given by the E and I firing rates $\nu_E$ and $\nu_I$ in the network:

$$
\begin{aligned}
E[I_{rec\ XI}] &= J_{XE}\nu_E - J_{XI}\nu_I \\
Var(I_{rec\ X}) &= J_{XE}j_{XE}\nu_E - J_{XI}j_{XI}\nu_I.
\end{aligned}
\tag{12}
$$

Where $J_{XY}$ is the population-wise connectivity defined by

$$
J_{XY} = j_{XY}p_{XY}N_Y.
\tag{13}
$$

We note that Eq 13 also defines the relation between the connectivity constants $j_{XY}$ of the spiking network and the connectivity weights $J_{XY}$ in the SSN model in Eq 3.

This leads to the system of self-consistent mean field network equations that arise from the $\Phi$ transfer function [16] and that need to be solved numerically

$$
\begin{cases}
\nu_E &= \Phi(\mu_E, \sigma_E) \\
\mu_E &= J_{EE}\nu_E - J_{EI}\nu_I + \mu_{ext} \\
\sigma_E^2 &= J_{EE}j_{EE}\nu_E + J_{EI}j_{EI}\nu_I + \sigma_{extE}^2 \\
\nu_I &= \Phi(\mu_I, \sigma_I) \\
\mu_I &= J_{IE}\nu_E - J_{II}\nu_I + r\mu_{ext} \\
\sigma_I^2 &= J_{IE}j_{IE}\nu_E + J_{II}j_{II}\nu_I + \sigma_{extI}^2.
\end{cases}
\tag{14}
$$

Since the $\Phi$ transfer function is derived for white noise input, this solution assumes that the Poissonian recurrent input $I_{rec}$ does not lead to large deviation to white noise. We refer to this approach as "Self-consistency solution" or $\Phi_{sc}$.

## Mapping LIF network—SSN

To meet the SSN activity regime with a simulation of spiking LIF neurons, we map the LIF network parameters to SSN parameters. The connectivity parameters $J_{XY}$ in the SSN correspond to the population-wise connectivity defined for the self-consistency solution $\Phi_{sc}$ according to Eq 13. The transfer function parameters $a$, $b$ and $n$ for each of the populations are obtained by fitting the F-I curve of the neuron obtained with the $\Phi$ function Eq 9, which depends on the LIF membrane time constant $\tau$, reset potential $V_R$, firing threshold $\Theta$, and the input noise $\sigma$ (Fig 1). The noise $\sigma$ is set to be the external noise $\sigma_{ext}$.

In LIF spiking networks, $\sigma_{ext}$ models the fluctuations in the membrane potential, which can be caused by fluctuations in the external network input as well as originate from intrinsical properties of the neuron [87]. We note that unlike in the Ricciardi mean-field solution $\Phi_{sc}$ (Eq 14), the SSN framework (Eq 3) does not explicitly model the input noise $\sigma$ to neurons embedded in a network. Instead, the effect of the noise is implicitly included in the power law approximation of the F-I curve. As a result, the noise in the SSN model is independent of the network activity leading to the assumption that the noise associated with recurrent input is negligible compared to the external noise $\sigma^2 = \sigma_{ext}^2 + \sigma_{rec}^2 \approx \sigma_{ext}^2$. This approximation holds for firing rates $\nu$ and connection strengths $j_{XY}$ in line with experimental observations [6, 33–40, 58, 59] (S1 Text).

## Supporting information

**S1 Fig. Additional map of computational regimes.** These maps are equivalent to the maps shown in Fig 5, and are generated for the connectivity of the supersaturating network shown in Fig 2 and the mouse V1 network shown in Fig 3A (A) The map shows many similarities to the map shown in Fig 5A. The balanced state is only defined for low $r$ values across external input values. The network can be inhibition stabilized for large input and low $r$, whereas supersaturation occurs for large input and high $r$. The supersaturation and ISN regions overlap. However, unlike Fig 5A, this network does not have a bistable regime in the range of inputs presented here. (B) Compared with the phase space in panel A, the ISN state (blue area) appears more difficult to achieve for this network as it requires much higher external input to reach. We show in Fig 3B that increasing $J_{EE}$ makes the ISN accessible for external inputs $\mu_{\text{ext}}$ lower than 100 mV/s.
(EPS)

**S2 Fig. Effect of network size on E/I input balance.** As the size of networks increases, the incoming excitatory and inhibitory inputs become more balanced. This is measured in E (blue) and I (red) neurons for the same bistability case as Figs 4 and 6C, at the point where the excitatory firing rate reaches 10 Hz (see inset above, for each network size). The balance factor measures the fraction of the excitatory input that is compensated by inhibition, $\frac{\mu_{XI}}{\mu_{XE}+\mu_{extX}}$. For $N = 4000$ neurons, E and I neurons receive nearly twice as much excitation as inhibition (BF $\approx$ 55% and 43% respectively) whereas for $N = 40000$, most excitatory input is cancelled by inhibition (BF $\approx$ 92% and 86% respectively). This shows that E-I balance gets tighter as the network size increases even though the firing rates are far from the balanced state limit (dashed line, inset) and the network response remains non-linear at low input level.
(EPS)

**S1 Text. Supplementary text with detailed information about the network models and parameters used in the main text.**
(ZIP)

# Acknowledgments

We thank Laura Bernáez Timón for comments on earlier versions of the manuscript. We thank Laura Busse, Julijana Gjorgjieva, Jochen Triesch, Simon Renner and all members of our group for scientific discussions. We thank Alexandra Vormberg and Andreas Nold for their code contribution at the early stage of the project.

# Author Contributions

**Conceptualization:** Nataliya Kraynyukova, Tatjana Tchumatchenko.

**Data curation:** Pierre Ekelmans.

**Formal analysis:** Pierre Ekelmans, Nataliya Kraynyukova.

**Funding acquisition:** Tatjana Tchumatchenko.

**Investigation:** Pierre Ekelmans.

**Methodology:** Pierre Ekelmans, Nataliya Kraynyukova, Tatjana Tchumatchenko.

**Resources:** Tatjana Tchumatchenko.

**Software:** Pierre Ekelmans.

**Supervision:** Nataliya Kraynyukova, Tatjana Tchumatchenko.

**Validation:** Pierre Ekelmans, Nataliya Kraynyukova, Tatjana Tchumatchenko.

**Visualization:** Pierre Ekelmans.

**Writing – original draft:** Pierre Ekelmans, Nataliya Kraynyukova, Tatjana Tchumatchenko.

**Writing – review & editing:** Pierre Ekelmans, Nataliya Kraynyukova, Tatjana Tchumatchenko.

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
