## [Decision Letter · Decision Letter 0]

19 Dec 2022

Dear Prof. Tchumatchenko,

Thank you very much for submitting your manuscript "Targeting operational regimes of interest in recurrent neural networks" for consideration at PLOS Computational Biology.

As with all papers reviewed by the journal, your manuscript was reviewed by members of the editorial board and by several independent reviewers. In light of the reviews (below this email), we would like to invite the resubmission of a significantly-revised version that takes into account the reviewers' comments.

We cannot make any decision about publication until we have seen the revised manuscript and your response to the reviewers' comments. You will see that there is significant disagreement among the reviewers about the suitability of this manuscript for publication.  However, two of the three reviewers raise crucial issues relating to novelty, implementation, and clarity and validity of explanations provided.  You will certainly need to address these issues thoroughly in the revision process.  Your revised manuscript will be sent to reviewers for further evaluation, to see if your revisions and responses convince them of its merits.

Sincerely,

Jonathan Rubin

Academic Editor

PLOS Computational Biology

Daniele Marinazzo

Section Editor

PLOS Computational Biology

Reviewer's Responses to Questions

**Comments to the Authors:**

Reviewer #1: I uploaded a pdf file

Reviewer #2: The manuscript presents an analysis of the stationary firing rate responses of a excitatory-inhibitory network of noisy leaky integrate-and-fire neurons as a function of constant external inputs. The stationary firing rates are obtained as the steady-state solutions (fixed points) of the mean-field equations (similar to Brunel J. Comp. Neurosci. 2000). Importantly, the transfer functions of single neurons ("f-I curves") that appear in the mean-field equations, are simplified by fitting a threshold-linear power-law function, which corresponds to the so-called stabilized supralinear network (SSN) model. The mapping to the SSN model enables a comprehensive mathematical analysis of the fixed points, and thus of the nonlinear firing rate responses to external stimuli, as a function of stimulus parameters and network connectivity. This analysis predicts a number of interesting and experimentally observed nonlinear responses (such as bistability, the paradoxical effect in inhibition stabilized networks and supersaturation), which may underlie important neural computations. The predicted responses are compared to simulations of the detailed spiking neural network and an excellent agreement is observed. Conversely, the paper can be seen as a demonstration that the highly tractable SSN model, for which many mathematical results exist, can be implemented with more realistic spiking neurons.

I found the manuscript interesting, sound and well-written. The theoretical approach via the SSN model is quite elegant. Furthermore, a theoretical understanding of the nonlinear response properties and neural computations in terms of biologically plausible spiking neural network model is an important current topic. It should also be noted that the model parameters (e.g. connectivity) have been chosen according to experimentally measured values from publicly available data. I have a few major concerns (see below), which should be relatively easy to address. Overall, I think that a revised manuscript that addresses all concerns listed below is suitable for publication in PLoS Computational Biology.

Major issues

1. When the effect of network size is investigated, the authors "re-scaled the recurrent connections J_{XY} by the factor 1/sqrt(N) as a function of network size N". I understand this sentence as follows: the weight at network size N is J_N=r*J/sqrt(N), where J is the weight before scaling (i.e. at N=4000) and r is some fixed factor of proportionality (from Table S2, I conclude r=sqrt{4000}). Now since J_N=j*p*N (Eq.13), the actual weight per synapse in the LIF network is j=(r*J/p)*N^(-3/2), i.e. the synaptic weights scale as 1/sqrt(N^3) and not as 1/sqrt(N) as in the limit for the balanced state theory. To obtain the desired balanced-state scaling (j~1/sqrt(N)), the rescaled weights should instead be J_N=J*sqrt(N)/r. If this latter scaling was indeed applied, then this is a minor issue and the authors should clarify the definition of the scaling more precisely using equations. Otherwise, Fig.6 and Fig.S2 should be redone with the correctly scaled weights (including new simulation) and the conclusions adapted if necessary. It may also explain why the authors did not observe a convergence to a balanced state in Fig.6. In this case, it would be interesting to know how large this network needs to be such that the balanced-state theory becomes acceptable.

2. A very similar analysis of the nonlinear response in LIF networks using the steady-state mean-field solutions has been recently performed by Sanzeni et al, Plos Comput Biol 2020 (ref.49). Given the similarity of the approach, the authors should clarify in more detail the difference and commonalities.

3. In many parts, it is stressed that the theory captures "medium-sized" spiking networks and the medium size appears to be a novel aspect of the paper. The "medium-size" theory is contrasted with the balanced-state theory, which is instead a "model for large or infinitely large networks". When reading the paper, I found the emphasis on medium size a little bit misleading because I expected a theory that captures finite size effects as in recent theoretical works. However, the present theory is just as well based on a mean-field limit of infinitely large networks as most other mean-field theories do, including the balanced-state theory; and as such it does not capture finite-size effects (e.g. fluctuation effects, correlations). Probably, the correct way to interpret their statement is as follows: there are different mean-field limits that lead to different limiting models and some are more useful than others in making predictions for the actual finite-size network at hand. Indeed, the SSN model is useful because the agreement with simulations of the medium-size LIF network is impressive, whereas the balanced-state theory fails to predict the responses of the medium-size network. The authors should make this rationale clear. They may also consider to replace "medium-sized" with the more innocent adjective "biologically-sized". Otherwise, the "medium-size" aspect could be confused with theoretical works on finite-size (or "mesoscopic") neural networks that deal specifically with the rather difficult problem of capturing finite-size effects such as fluctuations (e.g. Schwalger et al. Plos Comput Biol. 2017, Bressloff Phys Rev E 2010) or correlations (e.g. Tetzlaff et al. Plos Comput. Biol. 2012).

4. The balanced-state theory is also rejected based on the correlations between E and I input currents (Fig.2C, 4C, S2). The authors write, e.g. "If the network operates at balance, the E and I recurrent inputs will cancel out and lead to a correlation which approaches -1. Here, the E-I correlation [...] is close to zero, demonstrating that the network operates far from E-I balance". I am not sure whether this statement about strong negative correlations in the balanced state is true (unfortunately, no reference is given), and thus whether the conclusions are correct. In the balanced network model, neurons become independent as N->Infinity, and hence E-I current correlations should vanish, unless correlations are introduced externally (e.g. through shared external Poisson inputs as in Renart et al. Science 2010). However, in the present paper, there are no external correlations, and therefore the balanced state theory predicts 0 correlations (not -1), if I am not wrong. If I am correct, the authors should probably drop the corresponding figures.

Minor issues

- p.3 "the F-I curve of a LIF neuron given by the Ricciardi transfer function Φ" and Fig.1: The Richiardi transfer function assumes Gaussian white noise in the voltage dynamics of the LIF model but this assumption is not mentioned in the main text (it is mentioned in SI, though).

- Please provide the full expression for the input current I in Eq.8, including recurrent inputs and noise. Importantly, please provide equations for the noise in the main text, otherwise it is ambiguous what e.g. the parameter sigma is. In the case of Gaussian white noise, sigma^2 cannot be its variance (as written below Eq.8) and sigma cannot be its standard deviation (as written in SI) because the variance of white noise is infinity. Maybe call it noise strength or infinitesimal variance.

- Sec. "Noise contribution of exponential synapses" in SI is not very clear, esp. the derivation of the effective sigma for colored noise, which seems rather hacky. There are theoretical results on this problem, most importantly Alijani & Richardson (Phys Rev. E 2011), who have a nice rescaling of the Ricciardi function for colored noise (basically "take sigma of the effective model with Gaussian white noise such that the resulting var(V) equals the var(V) of the actual model with colored noise (both in absence of threshold). It will lead to similar firing rates for white and colored noise"). Another result on the firing rate for small tau_s has been derived by Brunel (Fourcaud-Trocme & Brunel Neural Comput 2002).

-p. 3 "the input to a neuron in a recurrent network [...] is equivalent to an Ornstein-Uhlenbeck process or white noise if the number of incoming PSPs is sufficiently large and the activity is irregular.": The condition of irregular activity is not precisely true. The white noise or OU process requires that the spiking activity is Poisson, i.e. it has no temporal correlations. Any stochastic spike train is, strictly speaking, irregular and bursty spiking is even more irregular than a Poisson process (CV, Fano factor > 1).

- Terminology "E-I balance": E.g., p.5 "To determine how close the network operates to E-I balance...": in this paper, the term "balance" is used in the rather narrow sense of the balanced-state theory of van Vreeswijk and co-workers. But E-I balance is more general, e.g. the model of Brunel (J. Comp Neurosci 2000) is also balanced if g=4 (perfect balance) or at least g>=4 and it has the same nonlinear response properties as in the present manuscript. Please provide a clear definition and terminology of "balance".

- p.5 reference to Eq.18: Eq.18 seems not to be in the main paper and probably needs reference to "Supporting Information".

- p.5 "correlation which approaches -1": In which limit? Maybe you mean 'correlation close to -1'? But see my point 4 above.

Typos:

======

-p.25 (SI), very last equation: equation for \\delta\\dot{\\nu}_I is wrong

-p.28 (SI): "for supersaturating networks (JEI /r − JII )": do you mean (JII/r-JEI<0)?

-p.12: misprint in last sentence of Results: \\rho_{E,I}?-1

-p.30: subscript "s" missing on tau_s in unnumbered equation

-p.30: equation for covariance: absolute value sign of |\\Delta t| is missing

Reviewer #3: In this manuscript, the authors use a least-squares fit to establish a mapping in firing rate space between E-I networks of LIF neuron models ("spiking networks") to two-dimensional rate models with power-law f-I curves ("SSN networks"). They verify that this mapping can predict firing rates and other dynamical properties of spiking networks using results from the considerably simpler SSN networks. In doing so, they make several useful observations about the the behavior and dynamics of spiking networks in various regimes.

The manuscript is well-written and contributes some useful insight and methods. I only have a handful of comments that should be addressed.

1) Abstract:

"Neural computations emerge from recurrent neural circuits that comprise hundreds to a few thousand neurons"

While it could be argued that "local" circuits in the cerebral cortex are restricted to approximately this size (even then, I'm not totally convinced, but I digress), the statement as written is too general. Please rephrase to clarify the context in which this statment is true (local circuits (or "a computational unit" eg column), cerebral cortex, rodent?).

2) "(LIF) model ... which represents an accurate description of cortical neurons both in vivo and in vitro"

This is a strong statement that many in the field would disagree with. It should be softened. The LIF can certainly be described as a useful approximation to real neurons or something similar, but I do not think that most computational neuroscientists (even those who use LIFs in the work) would agree with the current phrasing.

3) Somewhere in the paragraph(s) following Eq (1):

Please specify that a, b, and n were fit to LIF simulations. This would help the reader understand the approach.

4) In the paragraph following Eq (2):

Give an equation for mu_X.

5) Before Eq (3):

You explain that you will focus on the equilibrium states, but you are selling yourself short since you also consider the stability of such states. This is worth mentioning here to avoid giving the impression that stability will be ignored.

6) Page 11:

The balanced state limit is discussed and plotted in the figures, but an equation for the rates in this limit is not given (unless I missed it?). Please write the equation here or in Methods.

7) Page 11 "By considering the feedforward input before scaling":

I assume what is going on here is that external input is multiplied by sqrt(N) in the simulations (consistent with the balanced net approach), so it needs to be scaled by 1/sqrt(N) to say that rates converge to a fixed value. I think that many readers will not follow this reasoning, especially since I do not see the original sqrt(N) scaling of external input mentioned anywhere (unless I missed it). Please clarify that this scaling is used (if it is) and make this statement more clear. Note that this explanation might come naturally from writing an equation for rates in the balanced limit, as suggested above.

8) Last sentence before Discussion:

There appears to be a LaTeX rendering error in the parenthesis. I see an upside down question mark in the pdf.

9) Methods LIF spiking network:

The authors mentions the occurrence of synchronous spiking in some of their LIF simulations with delta synapses. While synchrony can be a real effect in such networks, it can also arise spuriously if the membrane potentials are updated immediately after each spike. Networks with exponential synapses do not have this problem. To avoid this problem in networks with delta synapses, the post-spike increments can be stored until after all membrane potentials have been updated (i.e., until after the forward Euler update if a forward Euler solver is used).

I do not know which approach the authors take, but if they increment membrane potentials immediately after each spike then they could consider modifying their code to perform post-spike increments after updating all of the membrane potentials to see if this modification avoids synchrony without resorting to exponential synapses.

10) Methods LIF spiking network:

Please give the values of time constant and delay parameters used for exponential synapses. Check that parameter values for all other parameters used in the study are also given somewhere.

**Have the authors made all data and (if applicable) computational code underlying the findings in their manuscript fully available?**

Reviewer #1: Yes

Reviewer #2: Yes

Reviewer #3: Yes

PLOS authors have the option to publish the peer review history of their article (what does this mean?). If published, this will include your full peer review and any attached files.

Reviewer #1: No

Reviewer #2: **Yes: **Tilo Schwalger

Reviewer #3: **Yes: **Robert Rosenbaum
---

## [Decision Letter · Decision Letter 1]

21 Mar 2023

Dear Prof. Tchumatchenko,

Thank you very much for submitting your manuscript "Targeting operational regimes of interest in recurrent neural networks" for consideration at PLOS Computational Biology. As with all papers reviewed by the journal, your manuscript was reviewed by members of the editorial board and by several independent reviewers. The reviewers appreciated the attention to an important topic. Based on the reviews, we are likely to accept this manuscript for publication, providing that you modify the manuscript according to the review recommendations.

Sincerely,

Jonathan Rubin

Academic Editor

PLOS Computational Biology

Daniele Marinazzo

Section Editor

PLOS Computational Biology

Reviewer's Responses to Questions

**Comments to the Authors:**

Reviewer #1: I thank the authors for their thorough responses to my questions, comments and criticisms. While I still am not 100% on board with all of the claims made here, what sort of world would it be if everyone agreed on everything? I would be happy to see the article be published in PLoS Comp Biol.

Reviewer #2: The authors have addressed all my concerns satisfactory.

Reviewer #3: The authors addressed all of my concerns with only one minor issue remaining. In response to one of my previous comments, they added a table of synaptic time constants and delay times (Table S3). The reported time constants are 200-250ms whereas synaptic timescales in cortical synapses are closer to 5-10ms. Similarly, the delay timescales are several orders of magnitude larger than realistic values for the size of networks studied in the manuscript. An unmyelinated axon can conduct a signal across 1 meter in 100ms, but the authors use delays several times longer than 100ms. Since the networks modeled in the manuscript are likely smaller than 1mm, these are highly unrealistic delays.

If the simulations work well (e.g., stability is achieved) with more realistic parameter values, then the simulations should be replaced with more realistic ones.

If the authors needed to choose their biologically unrealistic timescales to achieve stability in their simulations, this is a weakness of their study. It does not preclude the publication of the manuscript in PCB. All studies have weaknesses and, more specifically, nearly all modeling studies use some unrealistic parameter values. But the choice of parameter values should be mentioned and explained somewhere in the Results and/or Discussion.

**Have the authors made all data and (if applicable) computational code underlying the findings in their manuscript fully available?**

Reviewer #1: Yes

Reviewer #2: None

Reviewer #3: None

PLOS authors have the option to publish the peer review history of their article (what does this mean?). If published, this will include your full peer review and any attached files.

Reviewer #1: No

Reviewer #2: No

Reviewer #3: **Yes: **Robert Rosenbaum

Figure Files:

Data Requirements:

Reproducibility:

References:

---

## [Editor Report · Decision Letter 2]

11 Apr 2023

Dear Prof. Tchumatchenko,

We are pleased to inform you that your manuscript 'Targeting operational regimes of interest in recurrent neural networks' has been provisionally accepted for publication in PLOS Computational Biology.

Best regards,

Jonathan Rubin

Academic Editor

PLOS Computational Biology

Daniele Marinazzo

Section Editor

PLOS Computational Biology

---

## [Editor Report · Acceptance letter]

8 May 2023

PCOMPBIOL-D-22-01393R2 

Targeting operational regimes of interest in recurrent neural networks

Dear Dr Tchumatchenko,

I am pleased to inform you that your manuscript has been formally accepted for publication in PLOS Computational Biology. Your manuscript is now with our production department and you will be notified of the publication date in due course.

With kind regards,

Zsofia Freund
